# Correlations of the Gastric and Duodenal Microbiota with Histological, Endoscopic, and Symptomatic Gastritis

**DOI:** 10.3390/jcm8030312

**Published:** 2019-03-05

**Authors:** Hye Seung Han, Sun-Young Lee, Seo Young Oh, Hee Won Moon, Hyunseok Cho, Ji-Hoon Kim

**Affiliations:** 1Department of Pathology, Konkuk University School of Medicine, Seoul 05030, Korea; aphsh@kuh.ac.kr (H.S.H.); syoh@kuh.ac.kr (S.Y.O.); 2Department of Internal Medicine, Konkuk University School of Medicine, Seoul 05030, Korea; 3Department of Laboratory Medicine, Konkuk University School of Medicine, Seoul 05030, Korea; hanasys@kuh.ac.kr; 4R&D Center, BioCore. Co. Ltd., Seoul 08511, Korea; hscho@bio-core.com (H.C.); jhkim80@bio-core.com (J.-H.K.)

**Keywords:** microbiota, inflammation, endoscopy, stomach, duodenum

## Abstract

Mucosal inflammation is characterized by neutrophil and mononuclear cell infiltration. This study aimed to determine the gastric and duodenal microbiota associated with histological, endoscopic, and symptomatic gastritis. Dyspeptic adults who presented for evaluation were included. Subjects with either comorbidities or recent drug intake were excluded. Three endoscopic biopsies were obtained from the antrum, body, and duodenum. Next-generation sequencing for 16S ribosomal RNA V1–V2 hypervariable regions was performed. The correlation between the composition of microbiota and the degree of inflammatory cell infiltration, endoscopic findings, and Patient Assessment of Gastrointestinal Disorders Symptom Severity Index (PAGI-SYM) score was analyzed. In 98 included subjects, microbial communities in the antrum and body showed Bray–Curtis similarity; however, those in the duodenum showed dissimilarity. Histological and endoscopic gastritis was associated with the abundance of *Helicobacter pylori* and that of commensal bacteria in the stomach. The abundances of *Variovorax paradoxus* and *Porphyromonas gingivalis* were correlated with histological gastritis, but not with endoscopic or symptomatic gastritis. The total PAGI-SYM score showed a stronger correlation with the duodenal microbiota (*Prevotella nanceiensis* and *Alloprevotella rava*) than with the gastric microbiota (*H. pylori*, *Neisseria elongate*, and *Corynebacterium segmentosum*). Different correlations of the gastric and duodenal microbiota with histological, endoscopic, and symptomatic gastritis were observed for the first time at the species level. *H. pylori*-negative gastritis is not associated with endoscopic or symptomatic gastritis. Only *H. pylori*-induced endoscopic gastritis requires gastric cancer surveillance. Owing to the weak correlation with *H. pylori*, symptomatic gastritis should be assessed separately from histological and endoscopic gastritis.

## 1. Introduction

Gastritis remains challenging for clinicians, because symptoms may appear even in the absence of histological or endoscopic gastritis. As gastritis-related symptoms are nonspecific, it is thus difficult to detect *Helicobacter pylori*-infected subjects among the patients with functional dyspepsia [1]. Owing to the lack of difference between *H. pylori*-induced organic dyspepsia and functional dyspepsia, histopathological and endoscopic findings are more reliable for diagnosing *H. pylori* infection and estimating the risk of gastric cancer [2,3]. Furthermore, different clinical, morphological, and serological phenotypes have been reported between *H. pylori*-negative autoimmune gastritis (type A gastritis) and *H. pylori*-induced antral-dominant type B gastritis [4]. Aside from *H. pylori*, other species may induce gastritis; however, the contribution of other microbiota to symptom generation, mucosal inflammation, and endoscopic findings is still uncertain.

Next-generation sequencing (NGS) analysis of the 16S ribosomal RNA (rRNA) hypervariable regions has enabled the understanding of changes in the composition of the gastric microbiota induced by *H. pylori* infection [5,6]. Prior to the development of chronic gastritis, acute mucosal inflammation is characterized by simultaneous neutrophil and mononuclear cell infiltration [2,7]. Acute inflammation is often found when the mucosal microbiota is dominated by the pathogen such as *H. pylori* [5,6,8,9,10,11,12,13]. *H. pylori* affects adjacent organs, and is more abundant in the duodenal lumen than in the duodenal mucosa [14]. Because duodenal dysfunction and abnormal responses may also induce functional dyspepsia [15], the composition of the duodenal microbiota may differ between the symptomatic and asymptomatic subjects. Although recent advances in NGS analysis have enabled the identification of the composition of microbiota in the duodenal mucosa, microbiota other than *H. pylori* that contribute to mucosal inflammation, carcinogenesis, or symptom generation remain unknown.

A pathogenic species may induce mucosal inflammation, specific endoscopic findings, and gastrointestinal symptoms. Conversely, the microbiota that plays a protective role against inflammation might be underrepresented in the inflamed mucosa. In this study, we attempted to determine the composition of gastric and duodenal microbiota according to the degree of inflammatory cell infiltration and the presence of endoscopic gastritis. To understand the differences among histological, endoscopic, and symptomatic gastritis, we also analyzed the correlation between the composition of microbiota and gastrointestinal symptoms.

## 2. Materials and Methods

### 2.1. Study Subjects

Korean adults who visited our Digestive Disease Center for the evaluation of dyspeptic symptom were enrolled in a prospective setting. Subjects with either comorbidities or recent drug intake within the last 3 months (including antimicrobials, acid suppressants, antidepressants, antithrombotic agents, laxatives, lipid-lowering agents, hypoglycemic agents, probiotics, hormone replacement therapy drugs, and traditional Chinese medicine) were excluded. Ninety-eight subjects (mean age, 37.9 ± 11.9 years; 21 male and 77 female) provided signed informed consents prior to the procedure, and their details were omitted to ensure anonymity. None of the included subjects were vegetarians.

Ethics approval: This study was approved by the institutional review board of Konkuk University Medical Center which confirmed that the study followed the Declaration of Helsinki. This study is registered as KCT0000718 at the Clinical Research Information Service (https://cris.nih.go.kr).

### 2.2. Upper Gastrointestinal Endoscopy and Biopsy Procedures

Before endoscopic examination, subjects were asked to score their symptoms from 0 (none) to 5 (very severe) on the 20-item Patient Assessment of Gastrointestinal Disorders Symptom Severity Index (PAGI-SYM) questionnaire, as described previously [16]. Endoscopic procedures were performed by one gastroenterologist (Dr. S.-Y. Lee) using the GIF-Q260 endoscope (Olympus Co., Ltd., Tokyo, Japan). Three biopsies were obtained from the greater curvature sides of the mid-antrum and mid-body, in addition to the descending duodenum. To minimize the risk of bile contamination, duodenal biopsy specimens were obtained from the opposite side of the ampulla of Vater. Specimens were taken using 2.8 mm biopsy forceps (FB-230K or FB-24K-1, Olympus Co, Ltd., Tokyo, Japan).

Endoscopic findings were described according to the updated Sydney classification system in addition to *H. pylori*-related endoscopic findings [2,3,4,17,18]. The presence of the following was recorded: (1) multiple minute hemorrhagic spots in the fundus, (2) hypertrophic gastric rugae measuring over 5 mm, (3) mucosal nodularity, including small granular-type nodular gastritis (chicken skin-like mucosa showing multiple submucosal nodules measuring 1–2 mm in size), large nodular-type nodular gastritis (multiple submucosal nodules measuring 3–4 mm in size), and metaplastic gastritis (irregular whitish elevations and/or depressed patchy erythema), (4) advanced gastric atrophy (visible submucosal vessels extending up to the body from the antrum), (5) erosive gastritis (raised, regular-sized, hyperemic erosions), (6) chronic superficial gastritis (regular linear hyperemic streaks), (7) gastric xanthoma (yellowish plaque), and (8) hematin deposits (intramural hemorrhage). Other notable changes in the background mucosa were recorded.

### 2.3. Histopathological Assessment

Gastric and duodenal biopsy specimens were evaluated by one pathologist (Dr. H. S. Han) blinded to the subjects’ clinical information and status. All the biopsied specimens were fixed with 10% neutral-buffered formalin and were embedded in paraffin blocks. After hematoxylin-eosin (H&E) and Giemsa staining, microscopic findings were described. The specimens were observed with the aid of microscope (Olympus BX51, Olympus Co, Ltd., Tokyo, Japan). Degrees of neutrophil and mononuclear cell infiltration, atrophy, and intestinal metaplasia were scored from 0 to 3 (no, mild, moderate, and marked degrees) according to the updated Sydney classification system (Appendix A).

### 2.4. Library Preparation for 16S rRNA Gene Sequencing

The 16S rRNA V1–V2 hypervariable regions were analyzed based on recent study findings on the gastric and/or duodenal mucosal microbiota using endoscopic biopsied specimens [14,19,20]. To prepare the library for NGS analysis of the 16S rRNA V1–V2 hypervariable regions, DNA extraction was conducted for specimens obtained from the antrum, body, and duodenum, as described [21]. During DNA extraction, proteinase K (Takara Bio. Inc., Shiga, Japan) was used to effectively break down the thick bacterial cell wall, and incubation for cell lysis was performed at 100°C. All isolated samples were checked for quantity and quality prior to polymerase chain reaction (PCR) amplification.

Bacterial 16S rRNA V1–V2 hypervariable regions were amplified using primers (Meta-8F-V1, V2: AGAGTTTGATCMTGGCTCAG and Meta-338R-V1, V2: GCTGCCTCCCGTAGGAGT) [22]. PCR amplification of the V1–V2 hypervariable regions was performed in a thermal controller (TP600, Takara Bio Inc., Shiga, Japan) using 12 μL of DNA extract. The final PCR reaction mix volume (30 μL) was achieved by adding 15 μL of BioFact 2× multistar master mix (Biofact Co., Ltd., Daejeon, Korea) and 1.5 μL (10 pmole) of each primer. The initial denaturation was performed at 95 °C for 10 min. Thereafter, 40 cycles of denaturation at 95 °C for 30 s, annealing at 58 °C for 40 s, and extension at 72 °C for 5 min were carried out. The final extension was performed at 72 °C for 7 min. A negative control without template was included in PCR products.

End-repair, ligation, nick-repair, and final amplification processes for sequencing libraries were performed using Ion Plus Fragment Library Kit and Ion Xpress Barcode Adapters (Thermo Fisher Scientific Inc., Waltham, MA, USA) according to the manufacturer’s protocol. The 16S rRNA libraries were purified using Agencourt AMPure XP reagent (Beckman Coulter, Brea, CA, USA). The respective size of each library and the absence of contaminants were verified using high-sensitivity DNA Kit on Agilent 2100 Bioanalyzer (Agilent Technologies, Santa Clara, CA, USA).

### 2.5. Template Sequencing and Data Availability

Template preparation and sequencing were performed with the aid of Ion Chef System and Ion S5 XL System with Ion 530 Chip Kit (Thermo Fisher Scientific Inc., Waltham, MA, USA). The FASTQ data were generated using the Torrent Suite™ software version 5.8 (Thermo Fisher Scientific Inc., Waltham, MA, USA) by sequencing the V1–V2 hypervariable regions of 16S rRNA gene. The FASTQ sequence files are available in the National Center for Biotechnology Information (NCBI) Bioproject under the accession number of PRJNA486978.

### 2.6. Quality Control of Sequence Data

The FASTQ data were further analyzed for quality control. After obtaining the sequences from each sample, trimmed reads were mapped. The final files were generated by demultiplexing the data and removing the primer. Reliable data were obtained by excluding Phred quality scores below 20, polyclonal reads with low quality, and 3′-end adaptors of reads.

### 2.7. Bioinformatics Processing and Sequence Analysis

Ion Reporter™ software version 5.6 (Thermo Fisher Scientific Inc., Waltham, MA, USA) was used to process the FASTQ files. Read sequences were checked for a minimum read length of 150 bp. Data filtering was performed by removing low copy numbers (threshold <10) and reads with an alignment coverage of <90%. Based on Ion Reporter Metagenomics 16S algorithms, automatic analyses were conducted using the metagenomics workflow in the Ion Reporter software. The linked Quantitative Insights into Microbial Ecology (QIIME) workflow on the Ion Reporter Server processed the FASTQ files as input data, and reported operational taxonomic unit (OTU) tables as output data.

### 2.8. Screening and Assessment for Organisms

The sequence reads were aligned against the reference sequences exhibited in Greengenes version 13.5 (Greengenes Database Consortium) and MicroSEQ 16S rRNA Reference Library version 2013.1 (Thermo Fisher Scientific Inc., Waltham, MA, USA). With respect to taxonomy assignment, OTUs with ≥10 copy numbers were taxonomically classified using two databases. The reads were matched with clean OTUs showing a sequence similarity of >97% for the entire length, as described previously [23].

### 2.9. Assessment of Alpha and Beta Diversity Indices

To minimize the effects of uneven sampling, diversity indices were calculated after normalizing the read numbers in each sample. Alpha diversity was exhibited as Chao1, Shannon, and Simpson indices using the linked QIIME workflow. To determine beta diversity, transformed OTU counts were used for principal coordinate analysis (PCoA). A PCoA plot based on the Bray–Curtis dissimilarity was constructed using the linked QIIME workflow.

### 2.10. Statistical Analysis

For the comparison of PCoA plots, permutational multivariate analysis of variance test was performed using “vegan” R package version 2.2-1. For continuous variables, means and standard deviations were provided using the *t*-test. For continuous variables with asymmetrical distribution, median values and ranges were provided using the Kruskal–Wallis test. Categorical variables were described as frequencies using chi-square test or Fisher’s exact test. Analysis was performed using PASW version 17.0 (SPSS Inc., Chicago, IL, USA). A *p*-value of <0.05 was considered statistically significant.

For the comparison between the *H. pylori*-infected and non-infected subjects, >1% *H. pylori* relative abundance was defined as *H. pylori* infection [24]. For multiple comparisons, continuous variables were analyzed by using the analysis of variance (ANOVA) with Bonferroni correction. Chi-squared test with Bonferroni correction was used for categorical variables. Correlation analysis was performed to verify the relationship between the relative abundance of microbiota and the degree of inflammatory cell infiltration, and described as correlation coefficient values (*r*) with a *p*-value. Furthermore, correlation analysis was performed to evaluate the association between the relative abundance of microbiota and PAGI-SYM scores. To exclude correlations found by chance, statistical significance was set at *p* < 0.0083 (*p* < 0.05 divided by six symptom subscales) after multiple testing correction.

## 3. Results

### 3.1. Different Compositions Between the Gastric and Duodenal Microbiota

A total of 573 different species (10 phyla, 209 families, and 335 genus) were detected in the stomach in 98 subjects. In the duodenum, 500 different species (9 phyla, 183 families, and 260 genera) were found. Beta diversity indicated a significant similarity between the antrum and the body (Figure 1). Common species found in the antrum, body, and duodenum are summarized in Appendix A. Differences between men and women were found only in the relative abundance of *Prevotella pallens* (*p* = 0.009) and *Streptococcus* sp. (*p* = 0.007) in the stomach. The median relative abundance of *P. pallens* was higher in women (0.06%; range, 0–8%) than in men (0.02%; range, 0–11%). The median relative abundance of *Streptococcus* sp. was higher in men (0.09%; range, 0–6%) than in women (0.04%; range, 0–2%).

With respect to the relative abundance of species from different biopsy sites, the strongest correlation was observed between *H. pylori* in the antrum and *H. pylori* in the body (*r* = 0.904, *p* < 0.001). The correlation coefficient values for the relative abundance of *H. pylori* in the duodenum and *H. pylori* in the antrum and body were 0.064 (*p* = 0.528) and 0.260 (*p* = 0.010), respectively. Regardless of the presence of *H. pylori* infection, *Brevundimonas aurantiaca* was dominant in the duodenum in 89 (90.8%) subjects. Only 14 subjects showed *H. pylori* in the duodenum among the 31 *H. pylori*-infected subjects. *H. pylori*-infected subjects showed lower diversity indices in the stomach, and higher diversity indices in the duodenum than non-infected subjects (Table 1).

### 3.2. Gastric and Duodenal Microbiota Related to Mucosal Inflammation

The composition of the gastric microbiota differed according to the combined inflammation score (Figure 2A). The combined inflammation score was correlated with the relative abundance of three species in the stomach (*H. pylori, Variovorax paradoxus*, and *Porphyromonas gingivalis*) and two species in the duodenum (*H. pylori* and *Leptotrichia genomosp*).

The relative abundance of *Propionibacterium acnes* (*r* = −0.475), *Pseudomonas veronii* (*r* = −0.349), *Pseudomonas* sp. (*r* = −0.345), *Dechloromonas* sp. (*r* = −0.291), *Staphylococcus epidermidis* (*r* = −0.272), *Cloacibacterium rupense* (*r* = −0.267), *Escherichia coli* (*r* = −0.265), *Hydrogenophilus hirschi*, (*r* = −0.227), and *Bacillus* sp. (*r* = −0.225) was negatively associated with the combined inflammation score in the stomach. In the duodenum, the relative abundance of *Moraxella osloensis* (*r* = −0.297), *S. epidermidis* (*r* = −0.245), and *Actinomyces odontolyticus* (*r* = −0.217) was negatively correlated with inflammation. Gastric and duodenal microbiota that showed significant correlation with inflammatory cell infiltration at the genus or species level are summarized in Table 2.

### 3.3. Microbiota Associated with Abnormal Endoscopic Findings

*H. pylori* was more abundant in subjects with hemorrhagic spots (*n* = 6), hypertrophic rugae (*n* = 7), advanced atrophy (*n* = 9), and mucosal nodularity (*n* = 13) than in their counterparts (Figure 3). No correlation with the composition of microbiota was observed in subjects with other endoscopic findings (raised hyperemic erosions (*n* = 3), hematin deposits (*n* = 3), linear hyperemic streaks (*n* = 2), and gastric xanthoma (*n* = 1)). *P. acnes*, *P. veronii*, *Pseudomonas* sp., *S. epidermidis*, and *C. rupense* were abundant in the presence of regular arrangement of collecting venules, which indicates intact gastric mucosa without endoscopic gastritis (Appendix A). All species related to endoscopic gastritis (*H. pylori*, *P. acnes*, *P. veronii*, *Pseudomonas* sp., *S. epidermidis*, and *C. rupense*) were found among the species associated with histological gastritis.

### 3.4. Microbiota Associated with the PAGI-SYM Score

The total PAGI-SYM score showed positive correlations with *Prevotella nanceiensis* (*r* = 0.273) and *Alloprevotella rava* (*r* = 0.209) in the duodenum. Stronger correlations were found with the duodenal microbiota than the gastric microbiota (*H. pylori*, *r* = 0.165; *Neisseria elongata*, *r* = 0.143; *Corynebacterium segmentosum*, *r* = 0.143; *P. pallens*, *r* = −0.196; *P. acnes*, *r* = −0.171; *S. epidermidis*, *r* = −0.145). Correlation with the duodenal microbiota was mostly observed with bloating, nausea, vomiting, and lower abdominal pain (Table 3).

### 3.5. Association Between the Microbiota Correlated with Histological, Endoscopic, and Symptomatic Gastritis

In the presence of hemorrhagic spots on endoscopic examination, the severity scores of heartburn and regurgitation (*p* = 0.004), bloating (*p* = 0.001), and lower abdominal pain (*p* = 0.001) were significantly higher than those without hemorrhagic spots (Appendix A). Other endoscopic findings were not associated with symptom subscale severity.

### 3.6. Microbiota Associated with Gastritis in H. pylori-Negative Subjects

In 61 *H. pylori*-negative subjects, the combined inflammation score was correlated with the relative abundances of *V. paradoxus* (*r* = 0.670) and *P. gingivalis* (*r* = 0.259) in the stomach (Figure 2B). None of the gastric microbiota showed an inverse correlation with the combined score. In the duodenum, the combined inflammation score was inversely correlated with the abundances of *S. epidermidis* (*r* = −0.346) and *M. osloensis* (*r* = −0.305). No duodenal microbiota showed a positive correlation with the combined inflammation score.

Only eight subjects had specific endoscopic findings (hyperemic raised erosions in 3 subjects, hematin deposits in 3 subjects, and linear hyperemic streaks in 2 subjects). No significant correlation was found between the presence of these endoscopic findings and the composition of the microbiota.

The total PAGI-SYM score was correlated with the abundance *of N. elongata* (*r* = 0.207), *C. segmentosum* (*r* = 0.235), *P. pallens* (*r* = −0.237) in the stomach and that of *A. odontolyticus* (*r* = 0.243), *Pseudomonas grimontii* (*r* = 0.238), and *Paracoccus* sp. (*r* = −0.207) in the duodenum. Stronger positive correlations were found with the duodenal microbiota than with the gastric microbiota (Table 3).

### 3.7. Differences in Microbial Composition between Corpus- and Antrum-Dominant Atrophic Gastritis

A mild degree of atrophy was found in the antrum in 43 subjects and in the body in 36 subjects. Among these subjects, 20 showed a mild degree of atrophy in both the antrum and the body. Neither a moderate nor marked degree of atrophy was found in this study. The relative abundance of *H. pylori* (*p* = 0.001)*, M. osloensis* (*p* = 0.018), *P. acnes* (*p* = 0.011), and *Bacillus* sp. (*p* = 0.004) differed between the 23 antrum- and 16 corpus-dominant cases of atrophic gastritis. The median relative abundance of *H. pylori* (0.62% vs. 0%) was significantly higher in the antrum- than in the corpus-dominant gastritis cases. Conversely, *M. osloensis* (0.08% vs. 0.06%), *P. acnes* (9.20% vs. 3.20%), and *Bacillus* sp. (1.13% vs. 0.08%) were more abundant in the corpus- than in the antrum-dominant gastritis cases. Nevertheless, none of these species showed significant differences in their relative abundance between the 79 specimens with atrophy (43 antrum and 36 body) and the 117 specimens without atrophy (55 antrum and 62 body).

## 4. Discussion

In the present study, microbial communities in the duodenum showed dissimilarity with those in the antrum and body. The relative abundance of *H. pylori* in the stomach was associated with the most cases of histological and endoscopic gastritis, but only some cases of symptomatic gastritis. Only a weak correlation was observed between symptom scores and *H. pylori* abundance. Symptom scores showed stronger correlation with the duodenal microbiota than with the gastric microbiota. Therefore, avoiding an overgrowth of *H. pylori* may prevent most histological and endoscopic gastritis, but only some symptomatic gastritis. Furthermore, cases of *H. pylori*-negative gastritis involved abundant *V. paradoxus* and *P. gingivalis*. Such abundances were correlated with histological gastritis, but not with endoscopic or symptomatic gastritis.

Among the 573 gastric species detected in this study, the strongest correlation was observed between the degree of inflammatory cell infiltration and the relative abundance of *H. pylori*. *V. paradoxus* and *P. gingivalis* were also linked to inflammation; however, their correlation coefficient values and relative abundance were not comparable to those of *H. pylori*. Only *H. pylori*-dominant dysbiosis showed an extremely high relative abundance of up to 98.4% with significant changes on endoscopic and histological findings. These findings are additive to those of previous studies that reported microbial changes during *H. pylori*-induced chronic inflammation [5,6,8,9,10,11,12]. In those studies, diversity was decreased in *H. pylori*-dominant condition and was lower in the body than in the antrum. This supports our study findings that correlations between the combined inflammation score and species abundance of *V. paradoxus* and *P. gingivalis* are more prominent in the body than in the antrum. The impact of certain species seems to be low in a diverse environment such as that in the antrum.

Endoscopic and histological gastritis were associated with the relative abundance of *H. pylori* and those of *P. acnes, P. veronii*, *Pseudomonas* sp., *C. rupense*, and *S. epidermidis*. Discrepancy between the species related to histological and endoscopic gastritis was noted for *V. paradoxus* and *P. gingivalis*. The abundances of *V. paradoxus,* and *P. gingivalis* were correlated with histological gastritis, but not with endoscopic or symptomatic gastritis. The abnormal endoscopic findings observed in this study are consistent *H. pylori*-related endoscopic findings with an increased risk of gastric cancer. Intestinal metaplasia and gastric corpus atrophy increase the risk of intestinal-type gastric cancer, whereas hypertrophic rugae, diffuse redness, and nodularity increase the risk of diffuse-type gastric cancer [4,17,18,25,26,27].

*P. gingivalis* is a well-known periodontal pathogen associated with esophageal cancer [28]; however, it did not correlate with endoscopic findings that require gastric cancer surveillance [4,17,18]. This study also showed that the abundance of neither *V. paradoxus* nor *P. gingivalis* was associated with atrophy. Only the abundance of *H. pylori* was associated with antrum-dominant atrophic gastritis. These findings support that *H pylori*-negative gastritis does not progress to precancerous lesions [29].

In this study, the species associated with symptom scores were mostly inconsistent with those related to histological and endoscopic gastritis. A stronger correlation with the duodenal microbiota than with the gastric microbiota was observed only in symptomatic gastritis. The weak correlation between the total PAGI-SYM score and the abundance of *H. pylori* explains why other factors (i.e., female sex, young age, spicy food intake) are more correlated with symptoms than *H. pylori* [1,30,31,32]. Because the enteric nervous system (ENS) may be regulated by the inflammatory effects of the microbiota resulting in altered symptom sensitivity or cognitive function [33,34], our findings further suggest that the duodenal microbiota (*P. nanceiensis* and *A. rava*) and gastric microbiota (*H. pylori*, *N. elongata*, and *C. segmentosum*) may negatively affect ENS modulation via neurogenic inflammatory process. This increases understanding of the symptoms of *H. pylori*-negative subjects.

The present study has limitations. First, PAGI-SYM questionnaires were used instead of Rome criteria, because the validity of the Korean version of the Rome III questionnaires was shown to be low [35]. Nevertheless, we found that the species that were correlated with symptom severity differed from those related to histological or endoscopic gastritis. Second, most of the patients who visited the clinic were female subjects who wanted to be examined by a female gastroenterologist; thus, only 21 of our included 98 subjects were male. Because the composition of the microbiota did not differ between our male and female subjects, we assume that our study findings would not have changed by increasing the number of male subjects. Third, we could not confirm whether *S. epidermidis*, *P. acnes*, *P. veronii*, *Pseudomonas* sp., and *C. rupense* play a defensive role in reducing histological and endoscopic gastritis. Even in *H. pylori*-negative subjects, the degree of inflammatory cell infiltration increased with the relative abundance of pathogens (*V. paradoxus* and *P. gingivalis*), and none of the commensals demonstrated statistically significant differences. The beneficial role of commensal bacteria requires further investigation.

## 5. Conclusions

Different correlations of the gastric and duodenal microbiota with histological, endoscopic, and symptomatic gastritis were observed for the first time at the species level. Histological gastritis was associated with the relative abundances of *H. pylori*, *V. paradoxus,* and *P. gingivalis. H. pylori*-negative gastritis was not associated with endoscopic or symptomatic gastritis. *H. pylori* was the only pathogen associated with endoscopic gastritis, which requires gastric cancer surveillance.

Symptomatic gastritis should be evaluated and managed differently from histological and endoscopic gastritis, because it is more strongly correlated with the duodenal microbiota (*P. nanceiensis* and *A. rava*) than the gastric microbiota (*H. pylori*, *N. elongata*, and *C. segmentosum*). Thus, factors other than *H. pylori* infection status should be assessed in cases of symptomatic gastritis.

## Figures and Tables

**Figure 1 jcm-08-00312-f001:**
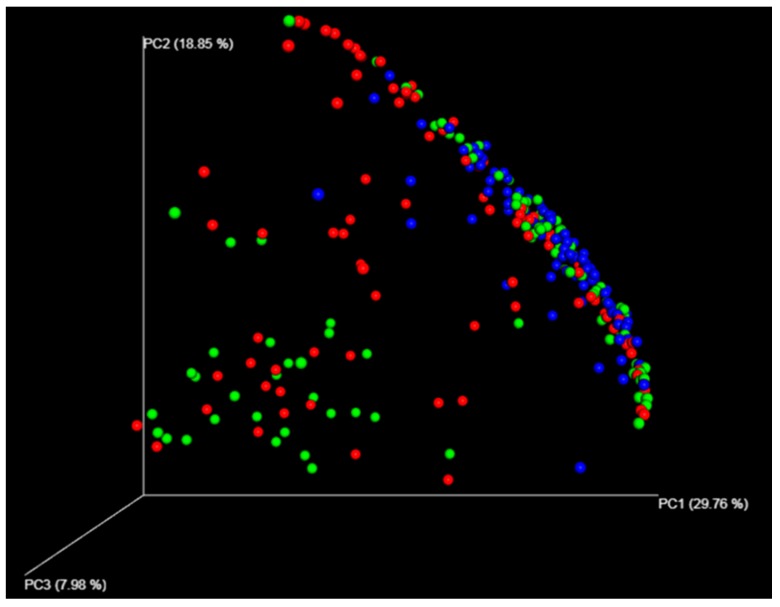
Beta diversity comparisons of microbial communities in the antrum, body, and duodenum. Principal coordinate analysis (PCoA) plots for the antrum (red), body (green), and duodenum (blue) are shown to determine Bray–Curtis distances. In the permutational multivariate analysis of variance, microbial communities in the antrum and the body showed similarity with a pseudo F-value of 4.52 (*p* = 0.003, *q* = 0.0030), whereas those in the antrum and duodenum showed dissimilarity with a pseudo F-value of 16.15 (*p* = 0.001, *q* = 0.0015). Furthermore, microbial communities in the body and duodenum showed Bray–Curtis dissimilarity with a pseudo F-value of 11.86 (*p* = 0.001, *q* = 0.0015).

**Figure 2 jcm-08-00312-f002:**
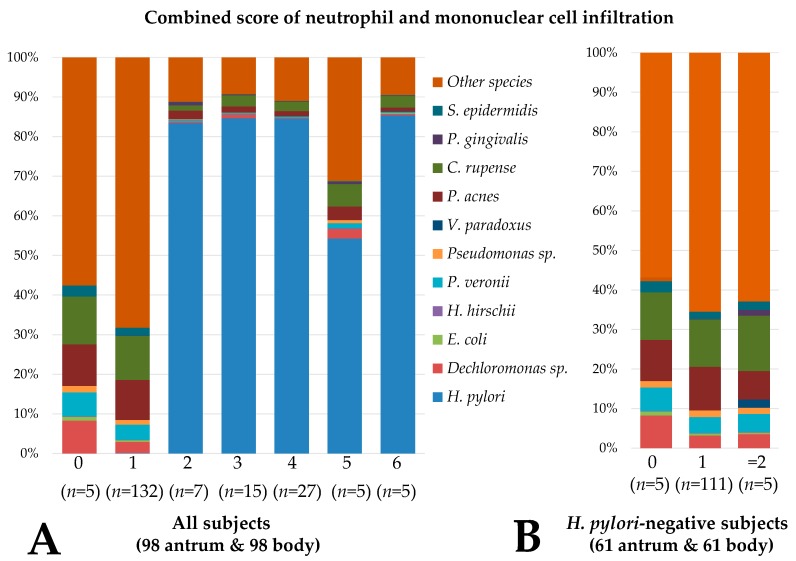
Composition of the gastric microbiota according to the combined inflammation score of neutrophil and mononuclear cell infiltration. (**A**) The composition of the gastric microbiota is shown according to the combined inflammation score. Scores of 0, 1, 2, and 3 indicate no, mild, moderate, and marked degrees, respectively, of neutrophil or mononuclear cell infiltration. The composition of gastric microbiota differed between the subjects with a score <2 and those with a score ≥2. (**B**) Among the 61 *H. pylori*-negative subjects, only five subjects had a combined inflammation score ≥2. Significant differences were found with the relative abundance of *Variovorax paradoxus* and *Porphyromonas gingivalis* between the subjects with scores ≥2 and those with scores <2. *S. epidermidis: Staphylococcus epidermidis; C. rupense: Cloacibacterium rupense; P. acnes: Propionibacterium acnés; P. veronii: Pseudomonas veronii; H. hirschii; Hydrogenophilus hirschii; E. coli: Escherichia coli*.

**Figure 3 jcm-08-00312-f003:**
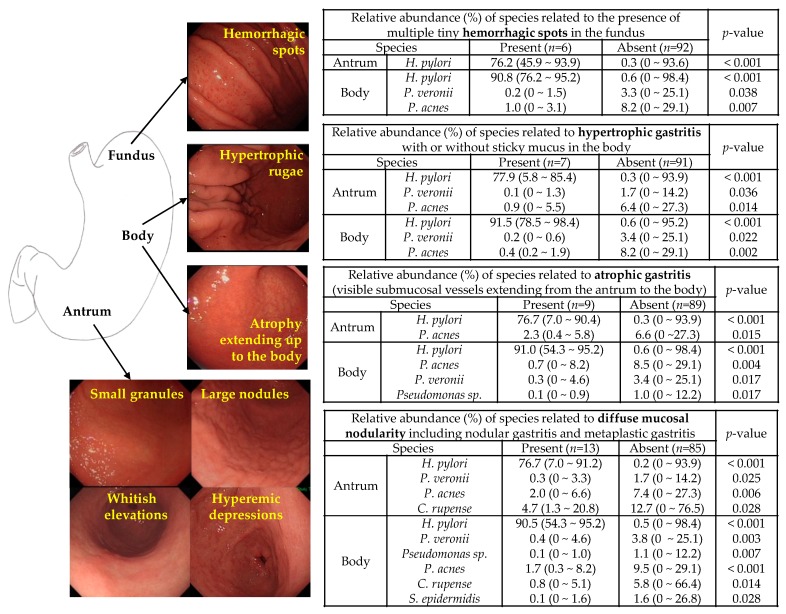
Significant endoscopic findings that were correlated with the relative abundance of microbiota. *Helicobacter pylori* was abundant in the presence of hemorrhagic spots, hypertrophic rugae, advanced atrophy, and mucosal nodularity, whereas *Pseudomonas veronii* and *Propionibacterium acnes* were abundant in the absence of these findings. *Pseudomonas* sp. was abundant in the absence of atrophy and nodularity. Moreover, *Cloacibacterium rupense* and *Staphylococcus epidermidis* were abundant in the absence of nodularity. The median relative abundance of each species were provided with range (minimum–maximum) using the Kruskal–Wallis test. *H. pylori: Helicobacter pylori; P. veronii: Pseudomonas veronii; P. acnes: Propionibacterium acnés; C. rupense: Cloacibacterium rupense; S. epidermidis: Staphylococcus epidermidis*.

**Table 1 jcm-08-00312-t001:** Differences between subjects according to the status of *Helicobacter pylori* infection.

Findings	With no *Helicobacter pylori* Infection (*n* = 61)	With *H. pylori* Infection (*n* = 31)	With past *H. pylori* Infection (*n* = 6)
Age (year-old)	35.4 ± 11.9	40.5 ± 9.9	49.8 ± 13.9 *^,^**
Sex (male:female)	16:45	3:28	2:4
Body mass index (kg/m^2^)	21.9 ± 3.4	22.4 ± 3.8	21.7 ± 2.2
Antrum: 16S ribosomal RNA (rRNA) sequencing analysis findings			
Target reads	20,759 ± 18373	18,744 ± 11,508	26,175 ± 9597
Chao 1 index	46.9 ± 16.6	46.7 ± 12.6	60.2 ± 9.4
Shannon diversity index	3.7 ± 0.5	3.4 ± 0.7 *	3.9 ± 0.3 **
Simpson diversity index	0.85 ± 0.05	0.78 ± 0.12 *	0.86 ± 0.04 **
Body: 16S rRNA sequencing analysis findings			
Target reads	14,998 ± 9734	13,041 ± 7148	12,271 ± 9245
Chao 1 index	39.2 ± 7.4	32.0 ± 8.9 *	33.8 ± 12.1
Shannon diversity index	3.4 ± 0.4	2.6 ± 0.6 *	3.3 ± 0.7 **
Simpson diversity index	0.82 ± 0.04	0.68 ± 0.13 *	0.82 ± 0.06 **
Duodenum: 16S rRNA sequencing analysis findings			
Target reads	9329 ± 4067	11,440 ± 6949	9834 ± 2278
Chao 1 index	31.5 ± 6.2	37.0 ± 7.2 *	32.8 ± 2.7
Shannon diversity index	3.2 ± 0.3	3.5 ± 0.4 *	3.3 ± 0.2
Simpson diversity index	0.81 ± 0.05	0.83 ± 0.05	0.83 ± 0.02
Antrum: updated Sydney system (no:mild:moderate:marked)			
Neutrophil	4:53:4:0	0:5:20:6 *	0:6:0:0 **
Mononuclear cell	58:2:0:1	6:10:12:3 *	6:0:0:0 **
Atrophy	37:24:0:0	15:16:0:0	3:3:0:0
Intestinal metaplasia	60:1:0:0	28:2:1:0	5:1:0:0
Body: updated Sydney system (no:mild:moderate:marked)			
Neutrophil	1:58:1:1	0:8:20:3 *	0:6:0:0 **
Mononuclear cell	59:0:1:1	6:6:18:1 *	6:0:0:0 **
Atrophy	35:26:0:0	23:8:0:0	4:2:0:0
Intestinal metaplasia	67:0:0:0	30:1:0:0	6:0:0:0
Duodenum: inflammatory cell infiltration (no:mild)			
Neutrophil	55:6	22:8	4:2
Mononuclear cell	9:52	4:27	0:6

For continuous variables, analysis of variance (ANOVA) with Bonferroni correction was used. For categorical variables, chi-squared test with Bonferroni correction was used. * Significantly different with 61 subjects with no *H. pylori* infection. ** Significantly different with 31 subjects with *H. pylori* infection.

**Table 2 jcm-08-00312-t002:** Significant correlations between the relative abundance of microbiota and the combined inflammation score.

Microbiota		Combined Inflammation Score	
Site	Level	Positive Correlation	Negative Correlation
Antrum	Genus	*Helicobacter* 0.794	*Dechloromonas* − 0.354
			*Staphylococcus* − 0.326
			*Pseudomonas* − 0.294
	Species	*Helicobacter pylori* 0.800	*Propionibacterium acnes* − 0.445
			*Pseudomonas veronii* − 0.360
			*Dechloromonas* sp. − 0.323
			*Cloacibacterium rupense* − 0.320
			*Pseudomonas* sp. − 0.320
			*Staphylococcus epidermidis* − 0.280
			*Hydrogenophilus hirschii* − 0.250
			*Streptococcus* sp. − 0.231
Body	Genus	*Helicobacter* 0.713 *Variovorax* 0.324	*Propionibacterium* − 0.469
			*Corynebacterium* − 0.457
			*Methylobacterium* − 0.454
			*Pseudomonas* − 0.402
			*Escherichia* − 0.342
			*Hydrogenophilus* − 0.307
			*Prevotella* − 0.307
			*Staphylococcus* − 0.280
			*Sphingomonas* − 0.258
			*Dechloromonas* sp. − 0.256
	Species	*Helicobacter pylori* 0.719 *Variovorax paradoxus* 0.325 *Porphyromonas gingivalis* 0.249	*Propionibacterium acnes* − 0.514
			*Pseudomonas* sp. − 0.377
			*Pseudomonas veronii* − 0.371
			*Escherichia coli* − 0.331
			*Dechloromonas* sp. − 0.279
			*Bacillus* sp. − 0.274
			*Staphylococcus epidermidis* − 0.265
Duodenum	Genus	*Helicobacter* 0.215	*–*
	Species	*Leptotrichia genomosp*. 0.218 *Helicobacter pylori* 0.201	*Moraxella osloensis* − 0.279
			*Staphylococcus epidermidis* − 0.245
			*Actinomyces odontolyticus* − 0.217

Pearson’s correlation coefficient (*r*) is shown for each microbiota.

**Table 3 jcm-08-00312-t003:** Gastric and duodenal microbiota correlated with the Patient Assessment of Gastrointestinal Disorders Symptom Severity Index (PAGI-SYM) scores.

PAGI-SYM Questionnaires	Site	Correlation with the Relative Abundance of Species	
		Positive Correlation	Negative Correlation
Heartburn and regurgitation	Stomach	*Corynebacterium segmentosum* 0.189	*Prevotella pallens* − 0.193
Bloating	Duodenum	*Prevotella nanceiensis* 0.283	*Propionibacterium acnes* − 0.220 *Prevotella pallens* − 0.215
	Stomach	*Helicobacter pylori* 0.239	-
Nausea and vomiting	Duodenum	*Prevotella nanceiensis* 0.302 *Actinomyces odontolyticus* 0.265	-
	Stomach	*H. pylori* 0.194	*Propionibacterium acnes* − 0.234 *Staphylococcus epidermidis* − 0.145
Upper abdominal pain	Stomach	-	-
	Duodenum	-	-
Fullness and early satiety	Stomach	-	*Dechloromonas* sp. − 0.189
Lower abdominal pain	Duodenum	*Prevotella nanceiensis* 0.313	-
	Stomach	*H. pylori* 0.228	
**PAGI-SYM Questionnaires**	**Site**	**Correlations Found in 61 *H. pylori*-Negative Subjects**	
		**Positive Correlation**	**Negative Correlation**
Heartburn & regurgitation	Stomach	*Corynebacterium segmentosum* 0.274	-
Bloating	Duodenum	*Pseudomonas grimontii* 0.308 *Cloacibacterium normanense* 0.281	-
Nausea and vomiting	Duodenum	*Actinomyces odontolyticus* 0.348 *Prevotella nanceiensis* 0.258	-
	Stomach	*Neisseria enlongata* 0.245	-
Upper abdominal pain	Duodenum	*Actinobacillus parahaemolyticus* 0.364 *Rothia mucilaginosa* 0.246 *Pseudomonas grimontii* 0.242	*Paracoccus* sp. − 0.242 *Neisseria perflava* − 0.242
Fullness and early satiety	Duodenum	*Porphyromonas catoniae* 0.332	-
	Stomach	*Pantoea* sp. 0.278	-
Lower abdominal pain	Stomach	*Neisseria enlongata* 0.249	-

Statistically significant microbiota are listed with Pearson’s correlation coefficient value (*r*). For each subscales, statistical significance was set at *p* < 0.0083 (*p* < 0.05 divided by six subscales) after multiple testing correction.

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
