# Peer review of "Correlations of the Gastric and Duodenal Microbiota with Histological, Endoscopic, and Symptomatic Gastritis"

_jcm, 2019, doi:10.3390/jcm8030312_

Reviewer 1 Report

I would suggest a minor spell check.

I would also recommend to extend the introduction in order to highlight the role of H.P. and other species and their correlation to the different gastritis "phenotypes".

Authors could extend "conclusions" or "discussion" in order to suggest potential utilities of their findings for the clinical practice.

Author Response

1. I would suggest a minor spell check.

à We apologize for the errors. The manuscript was checked again by the editing service after revision.

2. I would also recommend to extend the introduction in order to highlight the role of H.P. and other species and their correlation to the different gastritis "phenotypes".
à Thank you for your suggestion. After reading your comment, we added type A and type B gastritis in the first paragraph of the Introduction section (lines 40-42) as follow.

Different clinical, morphological, and serological phenotypes have been reported between H. pylori-negative autoimmune gastritis (type A gastritis) and H. pylori-induced antral-dominant type B gastritis [4].

For this, we also added our data on different microbial composition between the corpus-dominant and antrum-dominant atrophic gastritis in 3.7 (lines 289-300) as follow.

3.7. Differences in microbial composition between corpus- and antrum-dominant atrophic gastritis

A mild degree of atrophy was found in the antrum in 43 subjects and in the body in 36 subjects. Among these subjects, 20 showed a mild degree of atrophy in both the antrum and the body. Neither a moderate nor marked degree of atrophy was found in this study. The relative abundance of H. pylori (p = 0.001), M. osloensis (p = 0.018), P. acnes (p = 0.011), and Bacillus sp. (p = 0.004) differed between the 16 corpus- and 23 antrum-dominant cases of atrophic gastritis. The median relative abundance of H. pylori (0.62% vs. 0%) was significantly higher in the antrum- than in the corpus-dominant gastritis cases. Conversely, M. osloensis (0.08% vs. 0.06%), P. acnes (9.20% vs. 3.20%), and Bacillus sp. (1.13% vs. 0.08%) were more abundant in the corpus- than in the antrum-dominant gastritis cases. Nevertheless, none of these species showed significant differences in their relative abundance between the 79 specimens with atrophy (43 antrum and 36 body) and the 117 specimens without atrophy (55 antrum and 62 body).

3. Authors could extend "conclusions" or "discussion" in order to suggest potential utilities of their findings for the clinical practice.
à Thank you for your suggestion. We revised the Conclusions as follow (lines 363-370).

Histological gastritis was associated with the relative abundances of H. pylori, V. paradoxus, and P. gingivalis. H. pylori-negative gastritis was not associated with endoscopic or symptomatic gastritis. H. pylori was the only pathogen associated with endoscopic gastritis, which requires gastric cancer surveillance.

Symptomatic gastritis should be evaluated and managed differently from histological and endoscopic gastritis, because it is more strongly correlated with the duodenal microbiota (P. nanceiensis and A. rava) than the gastric microbiota (H. pylori, N. elongata, and C. segmentosum). Thus, factors other than H. pylori infection status should be assessed in cases of symptomatic gastritis.

Again, thank you for your helpful comments that strengthened our manuscript.

Reviewer 2 Report

Overall, there is data of interest to readers, but I think that certain phrasings, conclusions, and analyses may need to be altered. One of the main points I raise is that I think there should be an analysis of H pylori negative gastritis, and I think that would be of interest to readers. Another area of concern is the discrepancy between the results text and the figures/tables, where the text talks about combined inflammation scores, but the figures show separate neutrophil vs mononuclear inflammation. Thus, more consistency is needed.

The authors examine the microbiota of the stomach (antrum and body) and duodenum and correlate species found at these sites with histologic, endoscopic, and symptomatic gastritis. Assessing how the microbiota differs in various inflammatory states may point towards protective and pathogenic bacteria, and would improve our understanding of the microbiota-gut relationship in inflammation. Overall, I think that the data is of interest to readers, especially that certain bacterial species in the duodenum correlate with patient symptoms. The analyses and conclusions need to be altered, however, since certain questions remain or are raised. Below are some major and minor points.

1.       Few grammatical errors jump out, such as semi colons in the abstract instead of commas, and a period at the end of the abstract, but I am sure the editors can address these things better than I can. But to point out another error, reference 14 has the full title of the article cut off. Another one is line 47 has a lone closed parenthesis.

2.       In the methods for library preparation, prior to DNA extraction, were the samples fresh, formalin fixed, frozen, paraffin embedded, or something else?

3.       The overall study correlates inflammation and the microbiota, which the authors usually are careful in not pointing to the direction of causation. However, some instances the authors seem to suggest a backwards causation occurs. For example, in lines 46-47, the authors state “During acute inflammation, the mucosal microbiota is dominated by the pathogen such H pylori.” It is well known that H pylori leads to an acute inflammatory response, but this sentence does not phrase it as such, and seems to suggest a reverse causal relationship. Additionally, it is quite common to have acute gastritis without H pylori.

4.       Related to point #1, I think there may need to be separate analyses of patients with no H pylori and patients with H pylori. Acute gastritis occurs under many settings other than H pylori. The authors initially investigate these separate groups in Table 1, but then do not compare the microbiota differences in these separate groups in Figure 2. I think it would be of interest, and within the scope of the paper, to determine the microbiota as in Figure 2 but only in patients without H pylori. Alternatively, if H pylori is not considered, what would the distribution look like? This may be important in unmasking potential variations in the microbiota aside from, or independent of, H pylori. For example, in figure 2, the authors show the composition of the microbiota under various degrees of either neutrophilic or mononuclear infiltration. However, for some of these bars, it seems that the composition of the microbiota other than H pylori remains quite similar. That is, if we were to ignore H pylori in the inflamed mucosa, what would the distribution look like?

5.       For Figure 2, it is hard to see the colors and the text of the species. Since all 4 panels show the same species categories, perhaps just have one bigger key.

6.       For Figure 2, it is unclear what comparisons were made that are significant indicated by the stars. It seems like only the “mild” and “moderate” categories for neutrophils were compared, and only the “no” and “mild” categories for mononuclear infiltration. If that is the case, why not “no” vs “marked” and so on? It is unclear if those comparisons were done, which is possible since the legend indicates multiple comparisons. Please clarify.

7.       The authors show in Table 1 that there is a lower diversity index in patients with H pylori vs without. This finding is not surprising, given that the number of organisms of H pylori vastly outweigh the abundance of the other species. I am not familiar with these diversity indices, but a quick search states that both Shannon and Simpson diversity indices take abundance into account. Additionally, no discussion is included in regards to the diversity data. I think the decrease in diversity is likely due to the overwhelming number of H pylori organisms in the H pylori infected patients, but a discussion is warranted.

8.       Line 211, “degree” perhaps should be changed to “score.”

9.       From line 212 to 225, the authors talk about correlations between combined degrees (scores) of neutrophil and mononuclear cell infiltrations and bacterial species. However, only in Figure 3 is there some indication of combined scoring. Table 2, for example, separates neutrophil and mononuclear inflammation, but the text only talks about combined inflammation. Also, in the text (line 214-215), they mention the r values for the duodenum as H pylori r = 0.201 and Leptotrichia genomosp r = 0.218, but Table 2 shows different r values, likely because they are not combined scores in the table. The same goes for all the species that are negatively correlated (Lines 217-222). This makes the presentation of the results very confusing, and the table should reflect the highlighted results in the text (i.e. the combined scores). If the separated inflammatory scores are needed, they can be presented as a supplemental figure.

10.   The line beginning in 215 in reference to Figure 3 seems to come out of nowhere and is hard to follow. Please explain how a cutoff value of >4.75% was obtained, what that means, how it is related to AUC etc. Additionally, I (and therefore other readers) am not familiar with the graphs in Figure 3. It seems that cutoffs are determined by some pre-set values determined by arbitrary sensitivity and specificity thresholds? For example, does the analysis answer the question “What is the cutoff of abundance proportion such that it generates a sensitivity of greater than 70% and specificity greater than 50%?” I don’t know if that is how a cutoff value was determined, but that is what I am guessing. The methods section addressing this analysis also does not clarify it, in my opinion.

11.   The bacterial species beginning in 217 are not italicized, but I think should be.

12.   Table 2 formatting makes it difficult to read, for example the negative numbers seem to have separation between the negative sign and the number, and some words are cutoff due to formatting (for example, the word species is written as “specie” with the “s” being pushed to the next line. This is a minor formatting point.

13.   Figure 4, it seems that the values stated in the table within the figure are medians, but what is in the parentheses? I presume range? Please make this clear in the legend or in the table within the figure.

14.   Table 3: The authors divide the alpha value of significance of 0.05 by the 6 subscales, but have a table of testing 7 parameters, 6 subscales and the total score. I think it should be divided by 7 instead for this correction for multiple comparisons. Some say this Bonferroni type of correction may be too conservative, and perhaps a statistician can suggest a different type of correction.

15.   Lines 22, 276, 306, 335: I am not sure that histologic and endoscopic gastritides correlated with the “balance” of H pylori and commensals. Also, I think the phrasing is awkward, and it is not clear what it means to be correlated with the balance. Reading the paper, I can deduce it means that histologic and endoscopic gastritis positively correlated with H pylori, and negative correlated with some commensals. However, as I pointed out earlier, I think that since the abundance of H pylori automatically alters the % of commensals, these two variables seem dependent on each other, and thus perhaps it would be useful to ask what happens to the commensal proportions in the presence of H pylori as compared to the H pylori negative patients? In Figure 2, C rupense seems to occupy approximately 25% of the bacteria. However, when H pylori is added, this number decreases, as is expected just by adding a lot of an additional species. But if H pylori is removed from the data, is C rupense still at 25% of the remaining bacteria? If so, then I think that its negative correlation is less relevant. If the authors were to compare patients without H pylori with and without inflammation, perhaps C rupense is associated with less inflammation, to suggest a protective effect or that inflammation kills off C rupense. These possibilities should be discussed in the discussion. Again, I think it is important to categorize the inflammation as H pylori associated gastritis or H pylori not associated gastritis, as the authors began to do in Table 1.

16.   Section 3.5 beginning on Line 274: Many of these sentences are stated elsewhere or should be moved elsewhere or should be expanded. Line 276 seems more like a discussion point, and indeed is stated almost verbatim on lines 306  and 335; I would recommend deleting this line (276) and keeping it in the discussion/conclusion. Also, Line 277 seems more like a discussion point. Line 279 addresses what I have been bringing up, what does the non-H pylori gastritis look like? And the data to justify line 279 is not presented, which I think needs to be. The next line, Line 281, was already stated on Line 266, and thus can be deleted. The next line 282-283 can be stated under section 3.4, as it seems that this statement is relevant to the PAGI-SYM score. The next line starting with line 283 introduces the Supplementary Table 3 by stating that “Symptom scores were associated with the presence of hemorrhagic spots…,” but not all symptoms correlated with hemorrhagic spots. I would either list the ones that did, or say some symptom scores were associated with it.

17.   Supplementary Table 3 lists a table of 20x10 plus 2 extra values of neutrophil vs mononuclear inflammation, so it could be considered 20x12. The authors seem to use a Bonferroni correction by dividing the p value by 20 (as there are 20 symptoms tested). However, Bonferroni correction divides the p value by the number of hypotheses, which may be 240. I am not a statistician and am not qualified to evaluate this, but this may need some consultation. Either way, it does seem, as the authors suggest, that only hemorrhagic spots seem to correlate with symptoms.

18.   Lines 292-293: The presence of H pylori is known to cause a chronic active gastritis, which may be symptomatic. The findings in the study are consistent with this and I think this sentence can be rephrased as such, for example “The presence of H pylori was associated with histologic and endoscopic gastritis, as expected, but not with symptomatic gastritis. This is consistent with the fact that asymptomatic patients can have H pylori associated gastritis.”

19.   Line 297: V paradoxus and P gingivalis only seems to be linked to inflammation in the gastric body, and this should be noted and perhaps referenced to Table 2.

20.   Lines 304-305 regarding duodenal cancer: I think this is a stretch and there are many possible reasons why duodenal cancer is rare in H pylori induced gastric dysbiosis. I don’t think this suggestion is warranted, and if it is included would need a long exploration of why the discrepancy in microbiota may partially account for the lack of cancer in the duodenum, but I think that simply the lack of H pylori and inflammation in the duodenum in patients with H pylori associated gastritis is enough to explain why no duodenal cancer is seen in these patients. But other considerations may include the different type of mucin secreted (offering a protective effect), different immune microenvironment, and different epithelial cells that may be inherently more resistant to carcinogenesis.

21.   Line 306, I already addressed this somewhat, but it seems that gastritis correlates with H pylori, and possibly secondarily negatively correlates with these other species. But presentation of the data in a different way as I mentioned before is needed. It does not seem that its associated with the “balance” of these species, that would require far more testing. I think one would need to show H pylori patients with little inflammation also has increased C rupense for example as compared to H pylori patients with inflammation.

22.   Line 308-309: Again the association with P gingivalis and V paradoxus with gastritis, is this independent of H pylori? Do these organisms also cause inflammation by themselves or do they just proliferate when H pylori is present?

23.   Lines 313-315: “…endoscopic findings that require gastric cancer surveillance.” To my knowledge, there are no or maybe very rare endoscopic findings that require gastric cancer surveillance in my country, only histologic findings or genetic history. Can the authors elaborate which endoscopic findings require gastric cancer surveillance?

24.   Lines 215-216: I don’t think this statement is warranted with the data presented. One needs to explicitly look at H pylori negative gastritis, as I mentioned earlier, to have any support of this statement. But even if H pylori negative gastritis shows fewer endoscopic “risky” lesions, I don’t think that necessarily supports the statement. I don’t think the data presented in this study relates to H pylori negative gastritis and precancerous lesions. And we already know other conditions of H pylori negative gastritis that lead to precancerous lesions, such as autoimmune atrophic gastritis.

25.   End of the sentence in line 330, instead of “despite their presence” it should be “despite the presence of these potentially protective species,” or something to that effect. The “their” term is just a bit ambiguous within this sentence.

26.   Lines 335 and 338-340: As stated earlier, I don’t think “balance” should be used, and I don’t think a comment on duodenal cancer should be made.

Author Response

1. Few grammatical errors jump out, such as semi colons in the abstract instead of commas, and a period at the end of the abstract, but I am sure the editors can address these things better than I can. But to point out another error, reference 14 has the full title of the article cut off. Another one is line 47 has a lone closed parenthesis.

à Thank you for checking the details. We are wondering why commas have changed to semicolons in the abstract. We erased the semicolons and parenthesis. We also added title of reference 14.

2. In the methods for library preparation, prior to DNA extraction, were the samples fresh, formalin fixed, frozen, paraffin embedded, or something else?

à All the biopsied specimens were fixed with 10% neutral buffered formalin and were embedded in a paraffin block. We added this in Methods section (lines 100-101).

3. The overall study correlates inflammation and the microbiota, which the authors usually are careful in not pointing to the direction of causation. However, some instances the authors seem to suggest a backwards causation occurs. For example, in lines 46-47, the authors state “During acute inflammation, the mucosal microbiota is dominated by the pathogen such H pylori.” It is well known that H pylori leads to an acute inflammatory response, but this sentence does not phrase it as such, and seems to suggest a reverse causal relationship. Additionally, it is quite common to have acute gastritis without H pylori.

à Thanks for your concern. We revised the sentence as follow (lines 48-49).

Acute inflammation is often found when the mucosal microbiota is dominated by the pathogen such as H. pylori.

4. Related to point #1, I think there may need to be separate analyses of patients with no H pylori and patients with H pylori. Acute gastritis occurs under many settings other than H pylori. The authors initially investigate these separate groups in Table 1, but then do not compare the microbiota differences in these separate groups in Figure 2. I think it would be of interest, and within the scope of the paper, to determine the microbiota as in Figure 2 but only in patients without H pylori. Alternatively, if H pylori is not considered, what would the distribution look like? This may be important in unmasking potential variations in the microbiota aside from, or independent of, H pylori. For example, in figure 2, the authors show the composition of the microbiota under various degrees of either neutrophilic or mononuclear infiltration. However, for some of these bars, it seems that the composition of the microbiota other than H pylori remains quite similar. That is, if we were to ignore H pylori in the inflamed mucosa, what would the distribution look like?

à According to your suggestion, we revised Figure 2 based on the combined inflammation scores. We changed “the degree of neutrophil and mononuclear cell infiltration” to “combined inflammatory score” throughout the manuscript. We also added findings of the H. pylori-negative subjects in the Results section (lines 275-288) and in the bottom of Table 3.

5. For Figure 2, it is hard to see the colors and the text of the species. Since all 4 panels show the same species categories, perhaps just have one bigger key.

à Thanks for pointing out. For better resolution, we copied the bars directly from the original image in Figure 2.

6. For Figure 2, it is unclear what comparisons were made that are significant indicated by the stars. It seems like only the “mild” and “moderate” categories for neutrophils were compared, and only the “no” and “mild” categories for mononuclear infiltration. If that is the case, why not “no” vs “marked” and so on? It is unclear if those comparisons were done, which is possible since the legend indicates multiple comparisons. Please clarify.

à To make easier to understand, we erased insignificant findings from the figure legend and revised it as follow.

Figure 2. Composition of the gastric microbiota according to the combined inflammation score of neutrophil and mononuclear cell infiltration. (A) The composition of the gastric microbiota is shown according to the combined inflammation score. Scores of 0, 1, 2, and 3 indicate no, mild, moderate, and marked degrees, respectively, of neutrophil or mononuclear cell infiltration. The composition of gastric microbiota differed between the subjects with a score<2 and those with a score ≥2. (B) Among the 61 H. pylori-negative subjects, only 5 subjects had a combined inflammation score ≥2. Significant differences were found with the relative abundance of V. paradoxus and P. gingivalis between the subjects with scores ≥2 and those with scores<2.< p="">

7. The authors show in Table 1 that there is a lower diversity index in patients with H pylori vs without. This finding is not surprising, given that the number of organisms of H pylori vastly outweigh the abundance of the other species. I am not familiar with these diversity indices, but a quick search states that both Shannon and Simpson diversity indices take abundance into account. Additionally, no discussion is included in regards to the diversity data. I think the decrease in diversity is likely due to the overwhelming number of H pylori organisms in the H pylori infected patients, but a discussion is warranted.

à Thanks for your valuable comments. We agree with your opinion that decreased diversity is associated with the predominance of H. pylori. We discussed about this in the last part of the second paragraph in Discussion as follow (lines 317-322).

In those studies, diversity was decreased in H. pylori-dominant condition and was lower in the body than in the antrum. This supports our study findings that correlations between the combined inflammation score and species abundance of V. paradoxus and P. gingivalis are more prominent in the body than in the antrum. The impact of certain species seems to be low in a diverse environment such as that in the antrum.

8. Line 211, “degree” perhaps should be changed to “score.”

à We changed “combined degree” to “combined score” throughout the manuscript.

9. From line 212 to 225, the authors talk about correlations between combined degrees (scores) of neutrophil and mononuclear cell infiltrations and bacterial species. However, only in Figure 3 is there some indication of combined scoring. Table 2, for example, separates neutrophil and mononuclear inflammation, but the text only talks about combined inflammation. Also, in the text (line 214-215), they mention the r values for the duodenum as H pylori r = 0.201 and Leptotrichia genomosp r = 0.218, but Table 2 shows different r values, likely because they are not combined scores in the table. The same goes for all the species that are negatively correlated (Lines 217-222). This makes the presentation of the results very confusing, and the table should reflect the highlighted results in the text (i.e. the combined scores). If the separated inflammatory scores are needed, they can be presented as a supplemental figure.

à According to your suggestion, we deleted the findings on neutrophil and mononuclear cell infiltration score, and revised them into combined inflammation score.

10. The line beginning in 215 in reference to Figure 3 seems to come out of nowhere and is hard to follow. Please explain how a cutoff value of >4.75% was obtained, what that means, how it is related to AUC etc. Additionally, I (and therefore other readers) am not familiar with the graphs in Figure 3. It seems that cutoffs are determined by some pre-set values determined by arbitrary sensitivity and specificity thresholds? For example, does the analysis answer the question “What is the cutoff of abundance proportion such that it generates a sensitivity of greater than 70% and specificity greater than 50%?” I don’t know if that is how a cutoff value was determined, but that is what I am guessing. The methods section addressing this analysis also does not clarify it, in my opinion.

à We apologize for making you confused. To make the readers less confused, we erased Figure 3 with ROC curves from the manuscript.

11. The bacterial species beginning in 217 are not italicized, but I think should be.

à It seems that there was an error during the editing process. We revised the species in Italic font.

12. Table 2 formatting makes it difficult to read, for example the negative numbers seem to have separation between the negative sign and the number, and some words are cutoff due to formatting (for example, the word species is written as “specie” with the “s” being pushed to the next line. This is a minor formatting point.

à We corrected them by minimizing the contents in Table 2. Thanks for checking the details.

13. Figure 4, it seems that the values stated in the table within the figure are medians, but what is in the parentheses? I presume range? Please make this clear in the legend or in the table within the figure.

à We added in the figure legend (line 257) that it is range (minimum – maximum).

14. Table 3: The authors divide the alpha value of significance of 0.05 by the 6 subscales, but have a table of testing 7 parameters, 6 subscales and the total score. I think it should be divided by 7 instead for this correction for multiple comparisons. Some say this Bonferroni type of correction may be too conservative, and perhaps a statistician can suggest a different type of correction.

à According to the statistician’s recommendation, we removed the total score from the table and described it only in sentences (lines 260-263). Because the subscale scores are already included in the total score, we separated it form other 6 subscales.

15. Lines 22, 276, 306, 335: I am not sure that histologic and endoscopic gastritides correlated with the “balance” of H pylori and commensals. Also, I think the phrasing is awkward, and it is not clear what it means to be correlated with the balance. Reading the paper, I can deduce it means that histologic and endoscopic gastritis positively correlated with H pylori, and negative correlated with some commensals. However, as I pointed out earlier, I think that since the abundance of H pylori automatically alters the % of commensals, these two variables seem dependent on each other, and thus perhaps it would be useful to ask what happens to the commensal proportions in the presence of H pylori as compared to the H pylori negative patients? In Figure 2, C rupense seems to occupy approximately 25% of the bacteria. However, when H pylori is added, this number decreases, as is expected just by adding a lot of an additional species. But if H pylori is removed from the data, is C rupense still at 25% of the remaining bacteria? If so, then I think that its negative correlation is less relevant. If the authors were to compare patients without H pylori with and without inflammation, perhaps C rupense is associated with less inflammation, to suggest a protective effect or that inflammation kills off C rupense. These possibilities should be discussed in the discussion. Again, I think it is important to categorize the inflammation as H pylori associated gastritis or H pylori not associated gastritis, as the authors began to do in Table 1.

à After reading your comments, we erased the term “balance” throughout the manuscript. Furthermore, we added that there was no statistically significant correlation between the abundance of C. rupense and inflammation score in the H. pylori-negative subjects (lines 276-281). We also mentioned about this in the last part of Discussion s follow (lines 357-360).

Even in H. pylori-negative subjects, the degree of inflammatory cell infiltration increased with the relative abundance of pathogens (V. paradoxus and P. acnes), and none of the commensals demonstrated statistically significant differences. The beneficial role of S. epidermidis, P. acnes, P. veronii, Pseudomonas sp., and C. rupense requires further investigation.

16. Section 3.5 beginning on Line 274: Many of these sentences are stated elsewhere or should be moved elsewhere or should be expanded. Line 276 seems more like a discussion point, and indeed is stated almost verbatim on lines 306 and 335; I would recommend deleting this line (276) and keeping it in the discussion/conclusion. Also, Line 277 seems more like a discussion point. Line 279 addresses what I have been bringing up, what does the non-H pylori gastritis look like? And the data to justify line 279 is not presented, which I think needs to be. The next line, Line 281, was already stated on Line 266, and thus can be deleted. The next line 282-283 can be stated under section 3.4, as it seems that this statement is relevant to the PAGI-SYM score. The next line starting with line 283 introduces the Supplementary Table 3 by stating that “Symptom scores were associated with the presence of hemorrhagic spots…,” but not all symptoms correlated with hemorrhagic spots. I would either list the ones that did, or say some symptom scores were associated with it.

à We erased the sentences from Results section and moved some into Discussion section. With regard to Supplementary Table 3, we revised 20 questionnaires into subscales, and described the findings in lines 271-274.

In the presence of hemorrhagic spots on endoscopic examination, the severity scores of heartburn and regurgitation (p = 0.004), bloating (p = 0.001), and lower abdominal pain (p = 0.001) were significantly higher than those without hemorrhagic spots (Supplementary Table 3). Other endoscopic findings were not associated with symptom subscale severity.

17. Supplementary Table 3 lists a table of 20x10 plus 2 extra values of neutrophil vs mononuclear inflammation, so it could be considered 20x12. The authors seem to use a Bonferroni correction by dividing the p value by 20 (as there are 20 symptoms tested). However, Bonferroni correction divides the p value by the number of hypotheses, which may be 240. I am not a statistician and am not qualified to evaluate this, but this may need some consultation. Either way, it does seem, as the authors suggest, that only hemorrhagic spots seem to correlate with symptoms.

à According to the statistician’s recommendation, we changed 20 symptoms to 6 subscales, and erased the degree of neutrophil and mononuclear cell infiltration from the symptom table. Now, Supplementary Table 3 shows only the comparisons between the two groups (present vs. absent), which is easier for readers to understand.

18. Lines 292-293: The presence of H pylori is known to cause a chronic active gastritis, which may be symptomatic. The findings in the study are consistent with this and I think this sentence can be rephrased as such, for example “The presence of H pylori was associated with histologic and endoscopic gastritis, as expected, but not with symptomatic gastritis. This is consistent with the fact that asymptomatic patients can have H pylori associated gastritis.”

à We revised the sentences, as follow in lines 303-305.

The relative abundance of H. pylori in the stomach was associated with the most cases of histological and endoscopic gastritis, but only some cases of symptomatic gastritis. Only a weak correlation was observed between symptom scores and H. pylori abundance.

19. Line 297: V paradoxus and P gingivalis only seems to be linked to inflammation in the gastric body, and this should be noted and perhaps referenced to Table 2.

à We noted this in the Discussion section as follow (lines 319-322).

This supports our study findings that correlations between the combined inflammation score and species abundance of V. paradoxus and P. gingivalis are more prominent in the body than in the antrum. The impact of certain species seems to be low in a diverse environment such as that in the antrum.

20. Lines 304-305 regarding duodenal cancer: I think this is a stretch and there are many possible reasons why duodenal cancer is rare in H pylori induced gastric dysbiosis. I don’t think this suggestion is warranted, and if it is included would need a long exploration of why the discrepancy in microbiota may partially account for the lack of cancer in the duodenum, but I think that simply the lack of H pylori and inflammation in the duodenum in patients with H pylori associated gastritis is enough to explain why no duodenal cancer is seen in these patients. But other considerations may include the different type of mucin secreted (offering a protective effect), different immune microenvironment, and different epithelial cells that may be inherently more resistant to carcinogenesis.

à Thank you for your suggestion. We erased about duodenal cancer throughout the manuscript.

21. Line 306, I already addressed this somewhat, but it seems that gastritis correlates with H pylori, and possibly secondarily negatively correlates with these other species. But presentation of the data in a different way as I mentioned before is needed. It does not seem that its associated with the “balance” of these species, that would require far more testing. I think one would need to show H pylori patients with little inflammation also has increased C rupense for example as compared to H pylori patients with inflammation.

à We erased the term “balance”. Unfortunately, none of the commensal bacteria including C. rupense were significant in H. pylori-negative subjects. Thanks for your concern.

22. Line 308-309: Again the association with P gingivalis and V paradoxus with gastritis, is this independent of H pylori? Do these organisms also cause inflammation by themselves or do they just proliferate when H pylori is present?

à Yes, they were independent to H. pylori. We added the details in new section 3.6 and also in figure 2B.

23. Lines 313-315: “…endoscopic findings that require gastric cancer surveillance.” To my knowledge, there are no or maybe very rare endoscopic findings that require gastric cancer surveillance in my country, only histologic findings or genetic history. Can the authors elaborate which endoscopic findings require gastric cancer surveillance?

à To provide subjective criteria for H. pylori infection-related endoscopic findings with increased risk of gastric cancer, the Kyoto classification for gastritis was announced at the 85th annual meeting of the Japanese Society for Gastrointestinal Endoscopy in May 2013. The main contents focus on determining the gastric cancer risk by scoring the endoscopic findings of the background gastric mucosa as described in the references 4, 17, 18, 23, 24, and 25. Hypertrophic gastric folds, diffuse redness, and nodularity increase the risk of diffuse-type gastritis cancer, whereas intestinal metaplasia and advanced gastric atrophy increase the risk of intestinal-type gastric cancer. These important findings are mostly published in Japanese or in Korean language, so we added only few references in lines 328-331.

24. Lines 215-216: I don’t think this statement is warranted with the data presented. One needs to explicitly look at H pylori negative gastritis, as I mentioned earlier, to have any support of this statement. But even if H pylori negative gastritis shows fewer endoscopic “risky” lesions, I don’t think that necessarily supports the statement. I don’t think the data presented in this study relates to H pylori negative gastritis and precancerous lesions. And we already know other conditions of H pylori negative gastritis that lead to precancerous lesions, such as autoimmune atrophic gastritis.

à We erased 215-216 with Figure 3, and added sentences on H. pylori-negative gastritis in the Abstract, Introduction, Results, and Discussion. Also please refer to the revised Table 3 and Figure 2B.

25. End of the sentence in line 330, instead of “despite their presence” it should be “despite the presence of these potentially protective species,” or something to that effect. The “their” term is just a bit ambiguous within this sentence.

à We erased the sentence after erasing 4.75%, and revised it as follow (lines 360-363).

Even in H. pylori-negative subjects, the degree of inflammatory cell infiltration increased with the relative abundance of pathogens (V. paradoxus and P. gingivalis), and none of the commensals demonstrated statistically significant differences. The beneficial role of commensal bacteria requires further investigation.

26. Lines 335 and 338-340: As stated earlier, I don’t think “balance” should be used, and I don’t think a comment on duodenal cancer should be made.

à We erased descriptions on duodenal cancer and balance throughout the manuscript.

Again, thank you for your helpful comments and suggestions that strengthened our manuscript.

Reviewer 3 Report

The manuscript entitled “Correlation of the gastric and duodenal microbiota with histological, endoscopic, and symptomatic gastritis” is interesting and innovative . The Authors for the first time performed comprehensive studies concerning this subject. Nevertheless the Authors did not escape some ambiguities in the text. Suggestions of the reviewer are presented below:

Materials and methods

Is ethic approval for the study compatible with European Union law.

More accurate description of the patients included into the study must be added (age bracket, how many men and women have been included into the study). The information about mean age must be moved from the results chapter to material and methods chapter.

It is know that the diet may influence on the microbiota within the GI tract. Short information about diet of patients included into the study could be added (for example : is the diet similar in all persons or is there anyone vegetarian)

Specification and the size of biopsy forceps used in the study must be added

The name of microscope, under which the histopathological studies were performed must be added

Results

In table one the mean age of the particular group of patients must be added

the photos of histopathological examination would enrich the manuscript

In the opinion of the reviewer, figures 2 and 3 are too small, and the descriptions of these figures are difficult to read.

Discussion

It is commonly know that the enteric nervous system (ENS)  is one of the most important factor taking part in the inflammatory processes. Moreover the correlation between the ENS and microbiota are known. So the reviewer suggests to discuss the possibility roles of the ENS in results obtained during study.

Author Response

Materials and methods

1. Is ethic approval for the study compatible with European Union law?

à Thank you for the concern. In our country, IRB checks for “WMA Declaration of Helsinki – Ethical Principles for Medical Research Involving Human Subjects” instead of the European Union law. We added this in the last part of first paragraph in the Methods section under Ethics approval as follow (lines 74-76).

Ethics approval: This study was approved by the institutional review board of Konkuk University Medical Center which confirmed that the study followed the Declaration of Helsinki. This study is registered as KCT0000718 at the Clinical Research Information Service (https://cris.nih.go.kr).

2. More accurate description of the patients included into the study must be added (age bracket, how many men and women have been included into the study). The information about mean age must be moved from the results chapter to material and methods chapter.

à According to your recommendation, we moved this to the first paragraph of the Methods section (lines 70-71), and also added age and sex in Table 1. Because the corresponding author is a female gastroenterologist, about 70% of the visitors were female patients who wanted to be scoped by a female endoscopist at an academic medical center. We described about this in the Discussion section again as follow (lines 351-355).

Most of the patients who visited the clinic were female subjects who wanted to be examined by a female gastroenterologist; thus, only 21 of our included 98 subjects were male. Because the composition of the microbiota did not differ between our male and female subjects, we assume that our study findings would not have changed by increasing the number of male subjects.

As an evidence, we also analyzed whether there is a difference between men and women in the composition of microbiota, and added the findings in the first paragraph of 3.1 in the Results section as follow (lines 183-187).

Differences between men and women were found only in the relative abundance of Prevotella pallens (p = 0.009) and Streptococcus sp. (p = 0.007) in the stomach. The median relative abundance of P. pallens was higher in women (0.06%; range, 0%-8%) than in men (0.02%; range, 0%-11%). The median relative abundance of Streptococcus sp. was higher in men (0.09%; range, 0%-6%) than in women (0.04%; range, 0%-2%).

3. It is know that the diet may influence on the microbiota within the GI tract. Short information about diet of patients included into the study could be added (for example : is the diet similar in all persons or is there anyone vegetarian).

à Thank you for pointing out. Before the enrollment of the study, we conformed that all of the subjects are Koreans who enjoy Korean food every day. We added that there was no vegetarian in our subjects (lines 72-73).

4. Specification and the size of biopsy forceps used in the study must be added. The name of microscope, under which the histopathological studies were performed must be added.

à Thanks for your suggestion. We added about this in lines 84-85 as follow.

Specimens were taken using 2.8 mm biopsy forceps (FB-230K or FB-24K-1, Olympus Co, Ltd., Tokyo, Japan).

With regard to the name of microscope and the histopathological studies, we added it in lines 101-102 as follow.

The specimens were observed with the aid of microscope (Olympus BX51, Olympus Co, Ltd., Tokyo, Japan).

Results

5. In table one the mean age of the particular group of patients must be added. The photos of histopathological examination would enrich the manuscript.

à We added it in Table 1. The photos of histopathological examination are now added as Supplementary Figure 1 in lines 372-376.

Microscopic findings of the biopsy specimens. (A) In the antrum, the degree of neutrophil infiltration was none (score 0), while that of mononuclear cells was mild (score 1) (H & E stain, × 400). (B) The degrees of neutrophil and mononuclear cell infiltration in the antrum were marked (score 3); therefore, the combined inflammation score was 6 (H & E stain, × 400)

6. In the opinion of the reviewer, figures 2 and 3 are too small, and the descriptions of these figures are difficult to read.

à We apologize for poor resolution. Figure 2 is replaced with a bigger image. We erased Figure 3 after reading other reviewer’s comments.

Discussion

7. It is commonly know that the enteric nervous system (ENS) is one of the most important factor taking part in the inflammatory processes. Moreover the correlation between the ENS and microbiota are known. So the reviewer suggests to discuss the possibility roles of the ENS in results obtained during study.

à Thanks for your comments. We added sentences on ENS in last part of the 5th paragraph in the Discussion section as follow (lines 342-347).

Because the enteric nervous system (ENS) may be regulated by the inflammatory effects of the microbiota resulting in altered symptom sensitivity or cognitive function [31, 32], our findings further suggest that the duodenal microbiota (P. nanceiensis and A. rava) and gastric microbiota (H. pylori, N. elongata, and C. segmentosum) may negatively affect ENS modulation via neurogenic inflammatory process. This increases understanding of the symptoms of H. pylori-negative subjects.

Again, thank you for your valuable suggestions that strengthened our manuscript.

Round  2

Reviewer 2 Report

The authors addressed specific issues raised in the first review, but didn't apply those same concepts to other figures. For example, it is important to separate H pylori infected subjects from non H pylori infected subjects (such as in Fig 3). Additionally, new considerations are raised that I had not considered the first time.
